# Transmission Dynamics of Torque Teno Sus Virus 1 (TTSuV1) Between Wild and Farmed Pigs: A Molecular Tool for Monitoring Cross-Population Spillover in Swine

**DOI:** 10.3390/microorganisms13122751

**Published:** 2025-12-03

**Authors:** Xiaolong Li, Kristen N. Wilson, Carson W. Torhorst, Kayla Blake, Kim M. Pepin, Samantha M. Wisely

**Affiliations:** 1Department of Wildlife Ecology and Conservation, University of Florida, Gainesville, FL 32603, USA; xiaolong.li@ufl.edu (X.L.);; 2JBS Live Pork, Russellville, AR 72802, USA; 3National Wildlife Research Center, United States Department of Agriculture, Animal and Plant Health Inspection Service, Wildlife Services, Fort Collins, CO 80521, USA

**Keywords:** torque teno sus virus 1 (TTSuV1), swine population, molecular epidemiology, transmission inference, spillover, biosecurity

## Abstract

Torque teno sus virus 1 (TTSuV1) is a ubiquitous, non-pathogenic virus in both wild and domestic pigs and has potential value as a molecular marker for monitoring cross-population viral transmission and biosecurity breaches. In this study, we integrated genetic data with Bayesian transmission inference to examine the dynamics of TTSuV1 transmission within and between wild and farmed pigs in Arkansas. Transmission steps, defined as the number of transmission events linking two hosts, were used to identify barriers to transmission, such as biosecurity measures or geographic separation. TTSuV1 was highly prevalent in farmed pigs (96.59%) and invasive wild pigs (47.76%), with sequences from both groups distributed across four major subtypes. Transmission step analyses revealed that wild–farmed pig pairs were consistently separated by numerous transmission steps (>10 steps), indicating strong isolation and little evidence of recent spillover. In contrast, few transmission steps (≤5 steps) were common within populations, reflecting localized circulation. Our findings support the use of TTSuV1 as a molecular marker to characterize cross-population viral movement and spillover, as well as to delineate population structure in swine systems. Practically, this approach offers a novel framework for using viral genomics to detect cross-population spillover events and monitor biosecurity breaches in swine production systems.

## 1. Introduction

As an invasive species with a broad geographic range, wild pigs (*Sus scrofa*) harbor and transmit a wide spectrum of viral, bacterial, and parasitic pathogens that can threaten livestock and human health [1,2]. Their broad distribution across the southern United States has heightened concerns about the increasing risk of cross-population transmission, particularly to domestic livestock including swine, and dairy and beef cattle [3]. Wild pigs frequently overlap with agricultural lands, which creates opportunities for direct or indirect contact with farmed pigs at fence-line interactions, through shared water sources or contaminated stored feed, and because of improper carcass disposal on pig farms [4,5,6,7,8,9]. In the United States, swine production systems are highly diverse, ranging from small-scale family farms to large commercial operations. According to the United States Pork Industry 2023 Report, there were approximately 52,696 farms with fewer than 2000 head, 4573 farms with 2000~4999 head, and 3540 farms with 5000 or more head as of 2022 [10]. These data suggest that small and medium-sized farms still comprise the majority of pig operations nationwide, each with varying levels of biosecurity infrastructure and exposure risks. This interface between invasive wild pigs and diverse farm systems presents a significant biosecurity risk, as cross-population spillover could introduce novel or endemic pathogens into commercial pig farms. Understanding and monitoring these transmission pathways is critical for developing targeted surveillance strategies, mitigating biosecurity breaches, and safeguarding swine production systems.

Molecular epidemiology has become an essential tool for understanding how pathogens spread within and between populations [11,12]. By analyzing genetic differences among pathogen sequences sampled from infected individuals, researchers can reconstruct transmission histories and identify links between cases that may not be evident through traditional contact tracing [13,14]. This approach has been widely applied in human infectious disease studies, for example, in mapping the spread of HIV within social networks, or in tracing the global transmission of SARS-CoV-2 during the COVID-19 pandemic [15,16,17,18]. In wildlife systems, the application of molecular epidemiology has revealed pivotal transmission routes and clarified the ecological roles of animal reservoirs in the natural history of zoonotic pathogens such as rabies and swine influenza viruses [19,20,21]. To reconstruct the transmission history of a pathogen within or between populations, phylodynamic models usually need to infer transmission chains, which are sequential links between sampled infections that reflect who likely infected whom, based on genetic divergence and timing. A related concept, pairwise transmission steps, refers to the number of intermediate infections (transmission events) separating two individuals in a reconstructed transmission tree [22]. Fewer transmission steps suggest recent or localized pathogen spread among individuals, while a greater number of transmission steps implies indirect or distant connections. This framework is particularly effective when applied to fast-evolving pathogens that accumulate mutations rapidly and generate high levels of genetic diversity, which are features that enhance resolution in tracing transmission patterns.

Torque teno viruses (TTVs), a group of non-enveloped, single-stranded DNA viruses belonging to the family *Anelloviridae*, were first discovered in a human patient in 1997 [23]. Although usually associated with persistent, subclinical infections, human TTVs have been linked to various clinical conditions, including respiratory diseases, acute enteritis, viral hepatitis, and autoimmune rheumatic disorders [24]. Homologous viruses have been identified in a wide range of domestic and wild animal species, including swine, where they are known as torque teno sus viruses (TTSuVs). TTSuVs are classified into 2 genera, and TTSuV1 belongs to genus *Iotatorqueviruses* [25]. Although it is not considered pathogenic on its own, TTSuV1 has been associated with co-infections and is notable for its high prevalence, long-term persistence, and considerable genetic variability [26,27,28]. These characteristics make it an ideal candidate for studying long-term viral dynamics within and between swine populations. Previous studies have suggested the utility of TTSuV1 as a molecular marker to trace viral movement and infer contact networks in pig populations [29,30]. If further developed, such a tool could support passive surveillance of pathogen transmission, help identify inter-population spillover events and serve as a genomic indicator of farm-level biosecurity breaches.

Despite this potential, several critical knowledge gaps remain. While many studies report TTSuV1 prevalence and diversity, few have applied systematic phylogenetic or transmission modeling frameworks to understand its transmission dynamics. To date, no studies have combined dated phylogenies and statistical transmission inference to quantify how TTSuV1 moves within and between structured populations. Building on the concepts of transmission chains and pairwise transmission steps, an important next step is to determine the typical number of intermediate individuals separating infections in different host groups. For example, how many intermediate individuals typically appear in the transmission chains within and among wild and farmed pigs? Without empirical data on TTSuV1 transmission dynamics, its potential use as an indicator of viral movement or interpopulation contact in surveillance and risk assessment remains hypothetical.

In Arkansas, wild pigs are widely established across the state, posing challenges to both livestock production and environmental management [31,32]. Pig farming in the state, much like the national trend, includes a substantial proportion of small and medium-sized operations [10], which may face greater challenges in implementing and maintaining stringent biosecurity measures compared to large-scale commercial systems. Leveraging a partnership with the state wildlife agency and several commercial pig farms, we conducted a field-based molecular surveillance study to investigate TTSuV1 transmission dynamics among pig populations in Arkansas by collecting and analyzing viral DNA extracted from blood samples from both wild and domestic pigs across multiple locations in southwestern Arkansas.

Our objective was to apply transmission step analysis to identify population structures that act as barriers to pathogen transmission between wild and domestic pigs. Such barriers may include physical factors, such as biosecurity measures, and spatial factors, such as geographic distance. We hypothesized that numerous transmission steps would indicate isolation by biosecurity or by geographic distance, whereas few steps would reflect within-population transmission or, if occurring between host populations, potential spillover events. The integration of field sampling, genomic sequencing, and computational modeling provided a comprehensive framework for understanding TTSuV1 movement across different ecological contexts.

## 2. Materials and Methods

### 2.1. Sample Collection

Between 2024 and 2025, whole blood samples were collected from domestic and wild pig populations in Arkansas. We obtained domestic pig samples across five commercial pig farms, with one farm being the mutual source from where the other four farms receive gilts every 2–4 weeks. Besides movement of animals from the one source farm, no animals were traded or moved between the other four farms. Whole blood was collected from wild pig samples during routine animal control efforts conducted by the USDA Animal and Plant Health Inspection Service Wildlife Services. For farmed pigs, whole blood was collected by venipuncture on live animals undergoing routine veterinary care.

### 2.2. DNA Extraction and PCR Screening

Whole blood (3–5 uL) was placed directly on either Nobuto Blood Filter Strips (Advantec MFS, Inc., Tokyo, Japan) or Flinders Technology Associates (FTA) cards (QIAcard FTA Classic, Qiagen, Hilden, Germany) at the site of collection. To extract the total DNA, 1 mm square was cut from either substrate using flame sterilized scissors and forceps and placed in 900 µL of PureGene Red Blood Cell Lysis Solution (Qiagen, Germantown, MD, USA) and samples were incubated at room temperature for 30 min. After centrifuging the sample at 15,000× *g* for 2 min, the lysis solution was pipetted off and 600 µL of PureGene Cell Lysis Solution was added. The samples were then incubated at room temperature for three days. 20 µL of Proteinase K (MilliporeSigma Life Sciences Company, Burlington, MA, USA) were added to each sample and incubated for 24 h at 56 °C to digest protein. Following protein digestion, total DNA was extracted according to the manufacturer’s protocol. DNA was resuspended in 50 µL of PureGene DNA Hydration Solution (Qiagen, Germantown, MD, USA) and lightly vortexed and stored at room temperature for 24 h to allow the pelleted DNA to fully elute into solution. Final DNA product was stored at 4 °C until quantified.

DNA concentration was assessed using a NanoDrop One spectrophotometer (Thermo Fisher Scientific, Waltham, MA, USA). To minimize the risk of PCR inhibition, any sample with a DNA concentration above 100 ng/μL was diluted to 50 ng/μL. TTSuV1 was detected using a one-step PCR assay with previously published primers targeting a 678 bp genomic fragment covering parts of the untranslated region (UTR) and the open reading frame (ORF) 1, as well as the full ORF2 [33]. PCR products were visualized on 2% agarose gels to confirm amplicon size consistent with TTSuV1-positive samples.

### 2.3. Sequencing and Phylogenetic Analysis

PCR-positive samples were Sanger sequenced in both directions. Raw sequencing chromatograms were processed using Geneious Prime 2025.0.3 (https://www.geneious.com). Sequences were trimmed, quality-filtered, and aligned using the MUSCLE plugin within Geneious. Only high-quality consensus sequences were retained for phylogenetic analysis.

A neighbor-joining (NJ) phylogenetic tree was constructed using MEGA version 11 [34] with 1000 bootstrap replicates to assess branch support. Previously typed TTSuV1 sequences from the literature were also included into the phylogeny reconstruction as references to identify the types of viral sequences obtained in this study.

### 2.4. Temporal Signal Assessment and Dated Phylogeny Reconstruction

Before conducting molecular clock analysis, we assessed the presence and strength of the temporal signal in our sequence dataset using *TempEst* v1.5.3 [35]. Temporal signal refers to the correlation between genetic divergence and sampling time, a prerequisite for reliable time-calibrated phylogenetic inference. Without this signal, dated phylogenies can produce misleading or unstable estimates of divergence times. *TempEst* calculates root-to-tip divergence from a preliminary tree and evaluates its correlation with sample collection dates, providing insight into whether molecular evolution is measurable over the study’s time frame. We generated a maximum likelihood (ML) phylogenetic tree using *IQ-TREE* v2.2.2 [36] under an automatically selected substitution model and used this tree as input for *TempEst*. Root-to-tip divergence was plotted against sampling dates, and regression analysis was performed to assess the strength and direction of the temporal signal. Sequences that contributed to a negative or weak correlation were excluded from downstream time-scaled analyses to improve the robustness of the molecular clock model.

Due to the narrow sampling window of this study (spanning approximately one year), we applied a strict prior on the substitution rate during phylogenetic dating to prevent overestimation of rate variation. This prior was based on a previously reported substitution rate for TTSuV [28], and we used a normal distribution with a mean of 0.0005 substitutions per site per year and a standard deviation (sigma) of 0.00001 to constrain the evolutionary rate to a plausible range to avoid overfitting to short-term variation.

Dated phylogenetic analysis was conducted using *BEAST2* v2.7.3 [37]. To determine the most appropriate model combination, we performed model selection using path sampling to compare alternative demographic and clock models [38]. Specifically, we tested two coalescent population models, constant size and exponential growth, as well as two relaxed clock models, lognormal and exponential. Path sampling was run with 12 steps, each consisting of 10,000,000 MCMC iterations, and the first 50% of samples from each step were discarded as burn-in. Marginal likelihoods were calculated for each model pair, and the best-fitting combination was selected based on Bayes Factor comparisons [39].

The final analysis used the selected model combination and was run with a GTR + I substitution model, which was chosen based on model testing conducted in MEGA 11, and gamma-distributed rate heterogeneity among sites (four rate categories). The MCMC chain was run for 80,000,000 iterations to ensure adequate sampling and mixing. We assessed the convergence and effective sample sizes (ESS > 200) using *Tracer* v1.7.2 [40]. A maximum clade credibility (MCC) tree was summarized from the posterior distribution using *TreeAnnotator v2.7.5* [41], discarding the initial 10% of states as burn-in.

### 2.5. Transmission Tree Inference

We inferred the transmission tree using the R package TransPhylo v1.4.5, a Bayesian framework that reconstructs likely transmission events from a dated phylogeny by modeling both between-host transmission and within-host–pathogen evolution [42]. This approach accounts for unsampled individuals, variable infectious periods, and the uncertainty in phylogenetic timing and allows for probabilistic inference of who-infected-whom and the time at which transmission occurred.

The input for *TransPhylo* consisted of the MCC tree generated from *BEAST2* along with associated sampling times. Given that the generation time of TTSuV1 (i.e., the average interval between infection of a host and its subsequent transmission) had not been reported in the literature, we used porcine circovirus type 2 (PCV2), another circular, single-stranded DNA virus endemic to swine, as a proxy. Based on available data for PCV2 transmission dynamics [43], we modeled the generation time as a gamma distribution with a mean of 18 days. Inference of the transmission tree was carried out using 500,000 MCMC iterations, during which *TransPhylo* jointly estimated the transmission tree and several key epidemiological parameters, including the sampling proportion and the within-host coalescent rate. A burn-in of 10% was applied, and posterior distributions were monitored to ensure convergence and adequate sampling of the posterior space. We extracted the consensus transmission tree from the posterior distribution and used it in downstream analyses.

### 2.6. Transmission Step Quantification and Spatial Association Analysis

Pairwise transmission steps, defined as the number of inferred transmission events separating any two samples, were computed from the consensus transmission tree. This metric should not be confused with the number of nucleotide substitutions separating sequences in the phylogeny. While substitutions accumulate over evolutionary time, transmission steps reflect epidemiological links, i.e., the minimum number of unsampled individuals likely involved in the chain of transmission connecting two hosts. We generated pairwise matrices of step counts for all combinations of wild–wild and wild–farmed pig sample pairs.

We visualized the transmission step frequencies through histograms to compare distributions across different pair groupings (e.g., wild–wild, farm–farm, wild–farmed). Heatmaps of pairwise transmission steps were also created using the *pheatmap* package in R.

Geographic coordinates for sampling locations were used to calculate pairwise geographic distances between individuals. We then compared these distances with pairwise transmission steps that were taken from the clade-specific transmission trees to assess the association between geographic proximity and inferred transmission connectivity. Given that the source farm was geographically more distant from the wild pig sampling areas than the other farms, and its inclusion could have artificially inflated distances and distorted the trend line upward, we excluded the source farm from the wild–farmed subset.

Scatterplots were used to visualize the relationship between geographic distance and transmission steps. To test these associations statistically, we first used a Poisson regression model with pairwise transmission steps as the dependent variable and geographic distance as the predictor. However, inspection of model residuals and dispersion statistics indicated overdispersion in the data (dispersion parameter > 2). We therefore re-fitted the models using a negative binomial regression, which accounts for extra-Poisson variation and provides more reliable inference with overdispersed count data. Separate models were run for wild–wild pig pairs and wild–farmed pig pairs. Analyses were conducted using in-house R scripts.

## 3. Results

### 3.1. TTSuV1 Detection in Farmed and Wild Pigs in Arkansas

Between 2024 and 2025, we collected a total of 155 pig blood samples in Arkansas, and 117 of them tested positive for TTSuV1. Among the 88 samples collected from domestic pigs across 5 commercial farms, 85 were PCR-positive, yielding a prevalence of 96.59%. All blood samples from the source farm were TTSuV1-positive, and the prevalence ranged from 80% to 100% for the other four farms (Appendix A). For wild pig samples, the corresponding prevalence was 47.76% (32/67). These results indicated that TTSuV1 was widely present in both populations, with notably higher prevalence among farmed pigs.

### 3.2. Genetic Diversity of TTSuV1 in Both Populations

Of the 117 PCR-positive samples, 79 yielded high-quality viral sequences after sequencing and trimming and were retained for phylogenetic analysis. The neighbor-joining (NJ) tree grouped the sequences into four distinct clusters, corresponding to the four known subtypes of TTSuV1 (Figure 1).

Sequences from both wild and farmed pigs were distributed across all four subtypes, although some subtypes were more prevalent in one population than the other. Subtype 1c was the most frequently observed, appearing in both domestic and wild pigs across multiple locations, and subtype 1a was the least represented, with only a few sequences in each population.

### 3.3. Time-Scaled Phylogeny and Transmission Tree Inference

To ensure reconstruction of reliable dated phylogeny, we first assessed the temporal signal in our dataset. After removing 10 sequences that contributed to a negative or weak correlation, the resulting regression still showed limited temporal structure (R^2^ = 0.01; Appendix A). This further supported our choice of using a strict informative prior on the substitution rate, rather than relying on the molecular clock signal alone.

Model selection using path sampling in BEAST2 favored a relaxed molecular clock with lognormal distribution combined with a constant population size prior (Appendix A). This model was used for reconstructing dated phylogenetic trees from 69 sequences. The resulting MCC tree grouped sequences into four distinct clades, corresponding to the four TTSuV1 subtypes (Appendix A). Two of these clades, Clades 1 (corresponding to subtype 1c) and 3 (corresponding to subtype 1d), contained sufficient numbers of sequences for downstream transmission tree inference.

To reduce computational burden and improve interpretability, we performed transmission tree inference separately for each of the two selected clades. The consensus transmission trees revealed structured transmission patterns across both wild and farmed pig populations, with distinct chains of infection within each group (Figure 2). Direct transmission events (i.e., infector-infectee pairs) were inferred within farmed (n = 2 pairs) and wild (n = 3 pairs) populations, respectively; but none were found between wild and farmed pigs in either clade.

### 3.4. Transmission Steps Within and Between Populations

We investigated transmission dynamics within and between domestic and wild pig populations using TTSuV1 Clade 1 and Clade 3 separately. Pairwise transmission steps were calculated and summarized to compare patterns across population pair types within each clade (Figure 3, Table 1). Three types of pairwise relationships were examined: the number of transmission steps between all possible combinations of sample pairs within each clade, pairs between the source farm and other farms, and pairs between farmed and wild pigs. By observing the clustering of number of transmission steps in the histograms, we classified the transmission steps into four categories: direct transmission (1 step), few transmission steps (2–5 steps), moderate transmission steps (6–10 steps), and numerous transmission steps (>10 steps). We hypothesized that, in the absence of spillover between wild and domestic pigs, the minimum number of pairwise transmission steps would be low within wild or domestic populations but high between host populations. From these hypotheses, we inferred that few pairwise transmission steps (≤5) between wild and domestic pigs would indicate spillover events involving direct or recent transmissions.

Across both clades, the majority of pairwise comparisons fell into the numerous transmission step category, indicating long transmission histories between most sampled individuals. Observing all possible pairs within each clade, a small number of few-step events were observed. These included step counts below 5 and, in some cases, direct transmission between individuals from the same population, either wild–wild or farm–farm. These few step events suggest increased transmission within local clusters, particularly where viruses were sampled from pigs in close geographic proximity or in shared environments.

The distribution of steps among source farm–other farm pairs showed a moderately broad range. Some pairs were separated by fewer than 10 steps, but most still clustered above that threshold, suggesting limited transmission connectivity between farms. In contrast, all wild–farmed pig pairs were separated by more than 10 transmission steps in both clades. The minimum number of transmission steps between wild and farmed pigs was 13 and 23 for Clades 1 and 3, respectively, indicating consistently distant links between wild and domestic populations in the inferred transmission events.

For each clade, we visualized pairwise transmission steps using heatmaps (Figure 4). In Clade 1 (Figure 4A), the majority of transmission steps with few intermediate hosts (dark red) were observed within the wild pig population, indicating frequent within-group transmission. A limited number of transmission events with few steps (red) were also observed between the source farm and Farm B, between Farm A and Farm C, and between Farm B and Farm C. Given that the source farm was geographically distant from the other farms, and no animal movements occurred between farms after initial dispersal from the source, these between-farm transmission events likely originated within the source farm prior to pig transportation.

For Clade 3 (Figure 4B), sequences from the source farm dominated the clade and exhibited dense clusters of transmission links with few viral transmission steps between sampled pigs. Meanwhile, virus samples from Farms B and C and from wild pigs were more sparsely connected and generally showed numerous transmission steps between one another, as reflected by the lighter shading. Notably, no few or moderate number of transmission steps were found between pairs of wild and farmed pigs.

### 3.5. Spatial Association Between Transmission Steps and Geographic Distance

To evaluate whether spatial proximity was associated with viral transmission chains with few steps, we examined the relationship between geographic distance and pairwise number of transmission steps in two subsets: wild–wild pig pairs and wild–farmed pig pairs. We hypothesized that wild host pigs in close geographic proximity were more likely to have recent contact/connections, resulting in fewer inferred transmission steps, whereas greater geographic distances would typically correspond to older, less direct links leading to numerous transmission steps between pairs of viral samples. These patterns would reflect isolation by distance. In contrast, long transmission steps between geographically close hosts might instead suggest isolation driven by biosecurity or other non-spatial barriers.

Negative binomial regression results showed no statistically significant association between geographic distance and transmission steps for either wild–wild pairs (*p* = 0.12) or wild–farmed pairs (*p* = 0.99; Appendix A). These results indicate that, after accounting for overdispersion, the relationship between geographic distance and inferred pairwise transmission steps could not be distinguished from random variation. Scatter plots, however, revealed a positive relationship between geographic distance and transmission steps in the wild–wild pairs, suggesting that wild pigs which were closer together had fewer transmission events between them (Figure 5A). Although not statistically significant, this upward trend observed in wild–wild pairs suggests that wild pigs which were found near one another shared a virus genealogy with a more recent common ancestor than wild pigs that were farther apart and thus transmission events occurred more recently in wild pigs that were near one another than in wild pigs that were further apart from one another. In contrast, no apparent pattern was found between geographic distance and transmission steps in wild–farmed pairs (Figure 5B). Pairwise transmission steps between wild and farmed pigs were highly variable, with both numerous and few inferred transmission steps occurring between pairs of hosts at varying geographic distances. This lack of pattern suggests no clear spatial structure in the genealogy of evolution of the virus between wild and farmed pigs which we infer to mean that transmission between wild and farmed pigs had not occurred in the recent past.

## 4. Discussion

This study applied a molecular epidemiological framework to examine the transmission dynamics of TTSuV1 in wild and farmed pig populations in Arkansas. By integrating molecular viral detection and genetic sequencing, phylogenetic reconstruction, Bayesian transmission inference, and spatial analysis, we evaluated the potential of TTSuV1 as a molecular marker for monitoring cross-population transmission, spillover and biosecurity risks. Our main findings are threefold: (1) TTSuV1 was detected at a sufficient prevalence in farmed and wild pigs in Arkansas to conduct downstream molecular epidemiological analyses; (2) viral sequences grouped into four genetically distinct subtypes, with all subtypes found across both host populations present, although only two clades had sufficiently high sample sizes to conduct molecular epidemiological analyses; (3) inferred transmission trees revealed a distinct separation between farmed and wild pigs, with wild–farmed pairs consistently separated by longer transmission chains, suggesting no recent transmission, and isolation by biosecurity. In our inference, numerous transmission steps indicated barriers to transmission, whether physical (e.g., biosecurity measures) or ecological (e.g., geographic distance), whereas few steps suggested more recent transmission events. If a short transmission chain, i.e., few steps, were detected between wild and farmed pigs, this would be strong evidence of a recent spillover event, providing timely, actionable information for investigating and addressing potential biosecurity breaches. We also observed a positive trend between geographic distance and the number of transmission steps among wild–wild pairs, consistent with localized spread within social groups and isolation by distance, although this association was not statistically significant. These findings collectively underscore the value of TTSuV1 as a molecular marker to characterize cross-population viral movement and population structure in swine systems.

### 4.1. TTSuV1 Prevalence and Genetic Diversity in Swine Populations

We found a high prevalence of TTSuV1 in farmed pigs (96.59%) and a moderate prevalence in wild pigs (47.76%). These results are consistent with previous studies showing the widespread presence of TTSuV1 in both domestic and wild swine populations globally [27,28,33,44,45]. The higher prevalence among domestic pigs could be attributed to a combination of factors including greater population density, more frequent animal-to-animal contact, and sustained virus persistence in closed, intensively managed production environments [46,47,48]. These conditions can facilitate continual viral circulation and re-exposure of pigs to the virus, leading to higher detection rates.

Despite differences in prevalence, TTSuV1 sequences from both host populations were genetically diverse and distributed across four major subtypes. This genetic diversity enhances the utility of TTSuV1 as a tool for molecular epidemiological studies. Greater sequence variation provides more resolution when reconstructing phylogenetic relationships, which in turn improves the ability to trace transmission patterns and detect connections between individuals or populations. Subtype 1c was the most common, while subtype 1a was the least. Some subtypes were more prevalent in one population than the other, but no subtypes were exclusive to either population. This distribution pattern suggests that subtype structure is not rigidly confined to host type, emphasizing the broad circulation of TTSuV1 lineages across both wild and domestic pig populations.

### 4.2. Limited Cross-Population Transmission in the Current Period

Our transmission tree analyses based on time-scaled phylogenies revealed a clear separation in current transmission dynamics between wild and domestic pigs. Within each of the two well-sampled clades, all wild–farmed pig pairs of virus isolates were separated by more than 10 inferred transmission steps, whereas many wild–wild and farm–farm pairs were connected by fewer than five steps, and some by just one step which indicates direct transmission. In our interpretation, these long transmission chains between wild and farmed pigs reflect transmission barriers, whether maintained by biosecurity measures, ecological separation, or both. The lack of short transmission chains between the two groups suggests an absence of recent cross-population spillover in our dataset.

This finding is epidemiologically meaningful. It implies that under current conditions in Arkansas, wild pigs and farmed pigs are maintaining separate viral transmission networks. However, if few transmission steps were observed between wild and farmed pigs, it would point toward a recent breach in separation that should prompt targeted investigations to identify where and how contact occurred. Continued surveillance using viral markers like TTSuV1 could help detect any future breaches in farm biosecurity.

### 4.3. Spatial Transmission Dynamics Within and Across Populations

We observed a spatial pattern in wild pig transmission dynamics. Among wild–wild pig pairs, geographic distance tended to increase with the number of viral transmission steps, indicating localized viral spread consistent with the idea that hosts close together in space are more likely to have had recent contact and transmission. This pattern aligns with the social structure of wild pigs, which often form stable social groups known as sounders [49]. These social groups promote repeated contact and localized transmission, resulting in geographically clustered pairwise transmission steps. While this positive trend was not statistically significant, likely due to sample size limitations, the visual pattern in the scatterplot still suggests that proximity may facilitate more recent transmission events and that transmission step analysis may be a useful tool in elucidating contact networks among free-ranging animals. As a future direction, integrating viral data with movement tracking or observational data from devices such as camera traps to construct contact networks among wild pigs could provide a complementary view of transmission pathways. Comparing these empirical contact networks with phylogenetically inferred transmission trees would help assess how well genetic data capture real-world interactions and could reveal hidden or indirect routes of viral spread that might be missed through direct observation alone.

In contrast, we found no statistically significant relationship between geographic distance and transmission steps in wild–farmed pig pairs. This suggests that even when wild pigs and farms are located near each other, direct or recent viral transmission has not occurred. The lack of spatial structure in wild–farmed comparisons supports the interpretation that current transmission network structure occurs within rather than among the two populations.

To minimize distortion in this analysis, we excluded the source farm from the spatial dataset due to its much greater geographic distance from the wild pig sampling areas. Including this outlier would have artificially inflated distance values and skewed the fitted curves upward. Even without the outlier, wild–farmed pairs remained scattered across the step-distance space, reinforcing the conclusion that geographic proximity alone does not predict transmission linkage across populations. This absence of a spatial signal in viruses from farmed animals may reflect a combination of biological, behavioral, and management factors, including biosecurity infrastructure and low rates of interpopulation contact, affecting the risk of cross-population transmissions in swine.

### 4.4. Value of TTSuV1 as a Molecular Surveillance Tool

Our results affirm the potential of TTSuV1 as a molecular surveillance tool in swine epidemiology. Although non-pathogenic, its ubiquity, persistence, and genetic variability make it a useful proxy for understanding contact networks and transmission processes. In this study, TTSuV1 allowed us to detect population structure, measure connectivity, and identify areas of limited or absent cross-population mixing which are key features for monitoring disease risk and evaluating biosecurity.

TTSuV1 offers a low-cost, stable target for long-term surveillance. Because it is not under strong immune or vaccine pressure and persists in infected hosts, it may provide more consistent detection than transient pathogens. For example, TTSuV1 could be used to monitor viral flow between domestic and wild pig populations to detect potential biosecurity breaches. If sequences from wild pigs are observed closely linked to commercial farm sequences in the inferred transmission tree, that is, separated by only a few inferred transmission steps, this would suggest a point of contact. Upon detection, an inspection could be made to identify the source of contact and transmission such as shared access to feed sources, water runoff, fence-line interactions, or indirect exposure via contaminated materials. By identifying such short-chain transmission patterns, TTSuV1 analysis could help pinpoint the source and route of viral introduction, providing actionable information to investigate how and where contact occurred and to implement targeted mitigation strategies. This kind of forensic application would be especially useful in detecting and responding to breaches before more pathogenic viruses have a chance to spread.

In passive surveillance systems, TTSuV1 might be routinely sequenced from diagnostic or health-monitoring samples. Over time, building a phylogenetic database of sequences could reveal hidden transmission networks and validate suspected outbreak sources. For instance, if an outbreak of a more serious pathogen occurs on a farm, it would warrant a rapid switch from passive to active surveillance, enabling parallel genetic and epidemiological investigations of TTSuV1 to confirm the breach and identify the most likely introduction point based on existing transmission pathways, even if the primary pathogen is no longer detectable.

In summary, while TTSuV1 is not a pathogenic virus and does not warrant direct intervention, it serves as a valuable molecular marker for understanding transmission structure and identifying potential vulnerabilities in swine production systems. Its persistence, genetic diversity, and host ubiquity enable a wide range of surveillance applications that can complement traditional disease monitoring approaches.

### 4.5. Limitations and Future Directions

Several limitations of this study should be acknowledged. First, the short temporal sampling window (approximately one year) in our study limited the strength of the molecular clock signal, requiring reliance on external substitution rate priors. While the weak root-to-tip correlation supports this modeling choice, it also highlights the value of broader temporal sampling. Future studies spanning multiple years would allow for stronger time-calibrated inferences and better estimates of transmission timing.

Second, we inferred the transmission trees using a generation time distribution based on PCV2, due to the lack of published estimates for TTSuV1. While PCV2 is a biologically similar virus, we acknowledge that TTSuV1’s persistent, typically subclinical infection profile may involve longer or more variable transmission intervals. This limitation primarily affects the absolute temporal scaling of our inferred transmission trees, but not their relative structure. In other words, our results reliably capture the relative transmission relationships among hosts, whereas the exact timing and number of inferred transmission events depend on the assumed generation time. A longer generation time would compress the apparent number of transmission steps, and a shorter one would expand them, effectively shifting the boundaries that distinguish few, moderate, and numerous transmission step categories without changing their relative connectivity patterns. Therefore, our interpretations focused on comparing relative patterns of transmission rather than exact temporal estimates. Future empirical studies characterizing TTSuV1 shedding duration and transmission rates will be essential to refine these parameters and improve the quantitative resolution of transmission inference. Thirdly, while our sampling covered multiple farms and wild pig locations, the geographic scope was still limited. Broader regional sampling and integration with movement, land-use, and behavioral data would enhance the generalizability of our findings. In addition, although TTSuV1 was detected across all four major phylogenetic clades, only two clades had sufficient sequences for meaningful reconstruction of transmission trees. The high genetic diversity of TTSuV1 likely contributed to this uneven distribution of sequences across clades, leaving some clades with too few representatives for robust inference. Larger sample sizes would help overcome this imbalance, enabling more comprehensive evaluation of transmission dynamics across the full spectrum of viral diversity. Importantly, detecting rare spillover events may require much denser sampling than what is typically feasible in field studies. Future surveillance design may benefit from combining broad baseline sampling with intensified, targeted sampling near high-risk interfaces to better capture rare but epidemiologically meaningful spillover events.

Finally, the persistence of TTSuV1 outside of hosts remains poorly understood, and our sampling design, which relied on pig tissues and blood, captures only direct transmission among pigs. While this provides valuable insight into pig-to-pig contact networks, it does not fully represent other possible transmission routes that could connect wild and farmed populations, such as environmental contamination or indirect pathways mediated by humans or other animals. Pairing TTSuV1-based molecular surveillance with complementary systems, such as environmental sampling or ecological movement data, would help provide a more comprehensive picture of cross-population transmission risk.

## 5. Conclusions

This study demonstrates that TTSuV1 can serve as a practical molecular marker for investigating viral transmission structure in swine populations. By combining genetic data with phylogenetic, epidemiological, and spatial analyses, we showed that TTSuV1 circulation in Arkansas is currently structured within wild and farmed pig populations, with no evidence of recent cross-population spillover and spread. The spatial clustering observed in wild pigs, and the absence of spatial association in wild–farmed pairs, further reinforced this separation.

Although TTSuV1 is not a pathogenic virus, its persistence, high prevalence, and genetic variability support its use as a stable, informative marker for surveillance purposes. The application of TTSuV1 in molecular epidemiology can aid in detecting biosecurity breaches, reconstructing transmission routes, and identifying points of contact between wild and commercial pig populations. As demonstrated here, this approach provides a valuable foundation for data-driven biosecurity prioritization and can be adapted to other regions and viral systems for proactive disease monitoring and control.

## Figures and Tables

**Figure 1 microorganisms-13-02751-f001:**
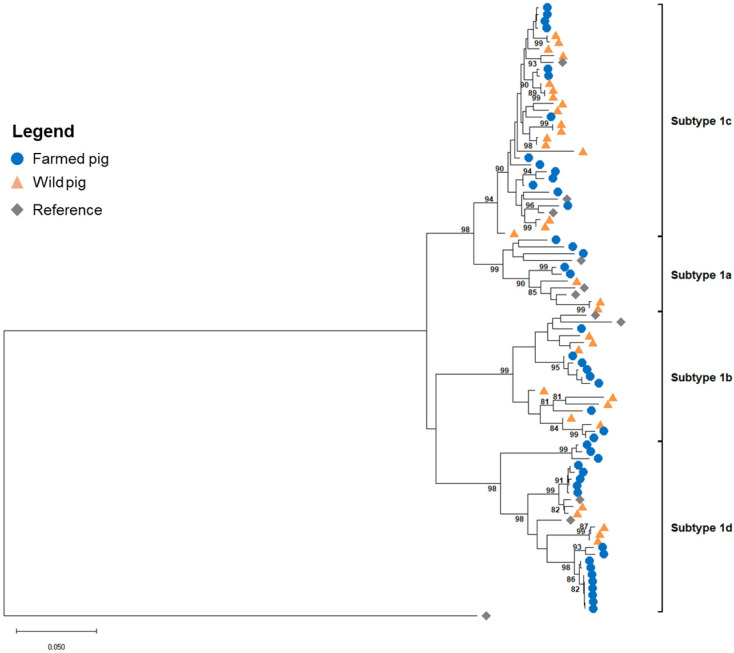
Neighbor-joining (NJ) phylogenetic tree of 79 TTSuV1 sequences obtained from wild and farmed pigs in Arkansas. Blue circles and orange triangles represent host population (wild vs. farmed). Eleven reference sequences (gray diamonds) with known subtypes were used (Appendix A). The tree was constructed with 1000 bootstrap replicates and was rooted with an outgroup sequence of TTSuV2. Support values > 80% are shown at major nodes. Scale bar represents the number of substitutions per site.

**Figure 2 microorganisms-13-02751-f002:**
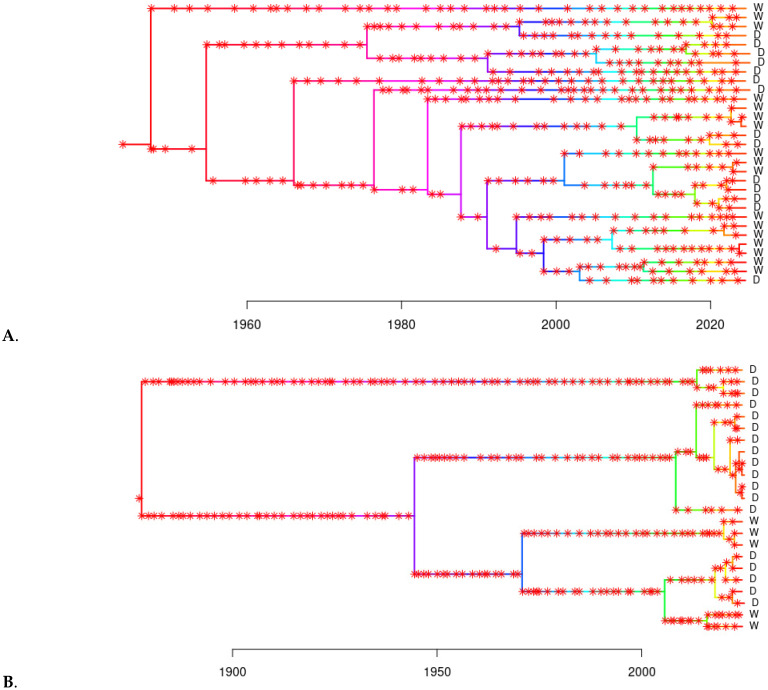
Consensus transmission trees inferred for Clade 1 (**A**) and Clade 3 (**B**) of the TTSuV1 time-dated tree. Each tree represents the most probable transmission history among sampled individuals within the clade, based on dated phylogenies and a Bayesian transmission model. Tips represent individual pigs sampled in this study, and asterisks represent inferred transmission events based on both within host evolution and evolution that occurred during transmission events. Tip labels are annotated with “D” for domestic pigs and “W” for wild pigs. Each tree was inferred independently based on the clade-specific MCC tree from BEAST2.

**Figure 3 microorganisms-13-02751-f003:**
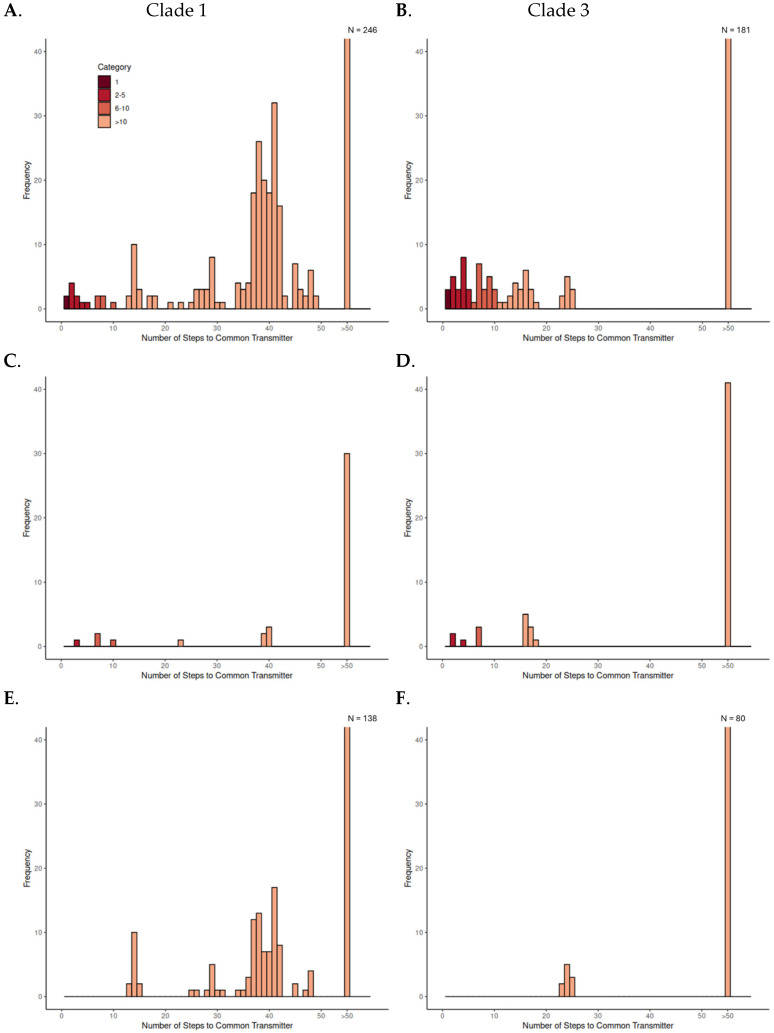
Histograms of inferred pairwise transmission steps between sample pairs within Clade 1 and Clade 3. Each panel shows the distribution of pairwise transmission steps: (**A**,**B**) all sample pairs within the clade; (**C**,**D**) pairs between the source farm and other farms; and (**E**,**F**) pairs between farmed and wild pigs. Bars are colored according to few, moderate, and numerous step categories. The final bin (“>50”) represents all pairs with more than 50 steps. The y-axis is truncated at 40 to enhance the visibility of smaller bars; the actual count of the “>50” bin exceeds the displayed maximum and is indicated above the bar (e.g., N = 246).

**Figure 4 microorganisms-13-02751-f004:**
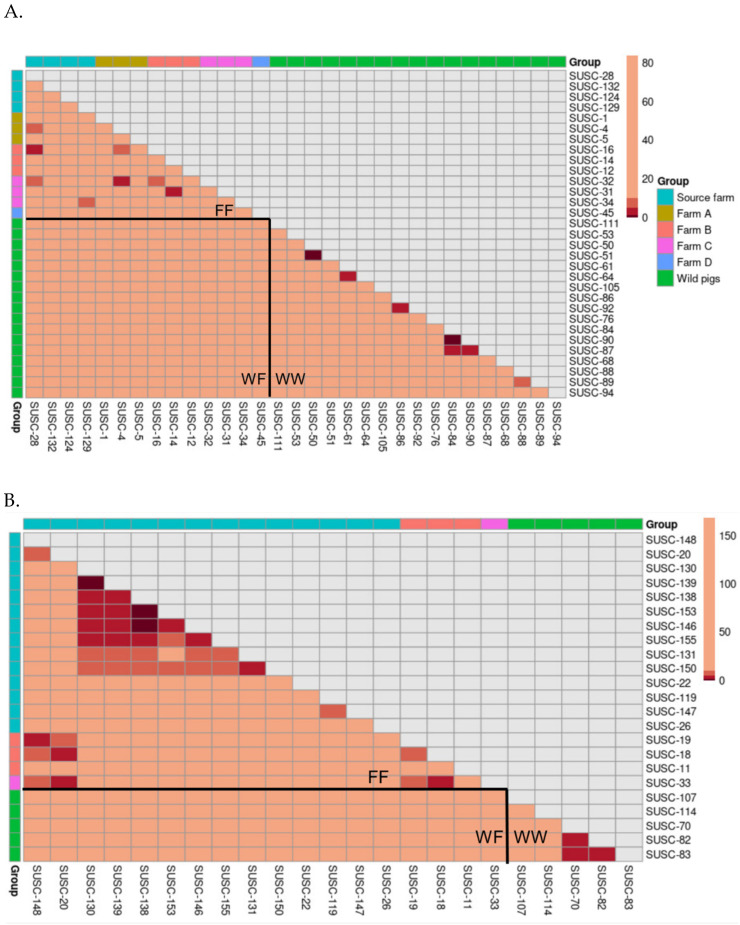
Heatmaps of pairwise transmission steps between individuals in two major clades. (**A**): Clade 1; (**B**): Clade 3. Cells are colored according to step intervals: 1 step (dark red), 2–5 steps (red), 6–10 steps (orange), and more than 10 steps (light beige). Tip labels are ordered by group. The color legend reflects these categories for easier interpretation of the transmission structure. The upper triangle of the symmetric matrix was masked for clarity, and the lower triangle was divided into three sections: farm–farm (FF) pairs, wild–farmed (WF) pairs, and wild–wild (WW) pairs.

**Figure 5 microorganisms-13-02751-f005:**
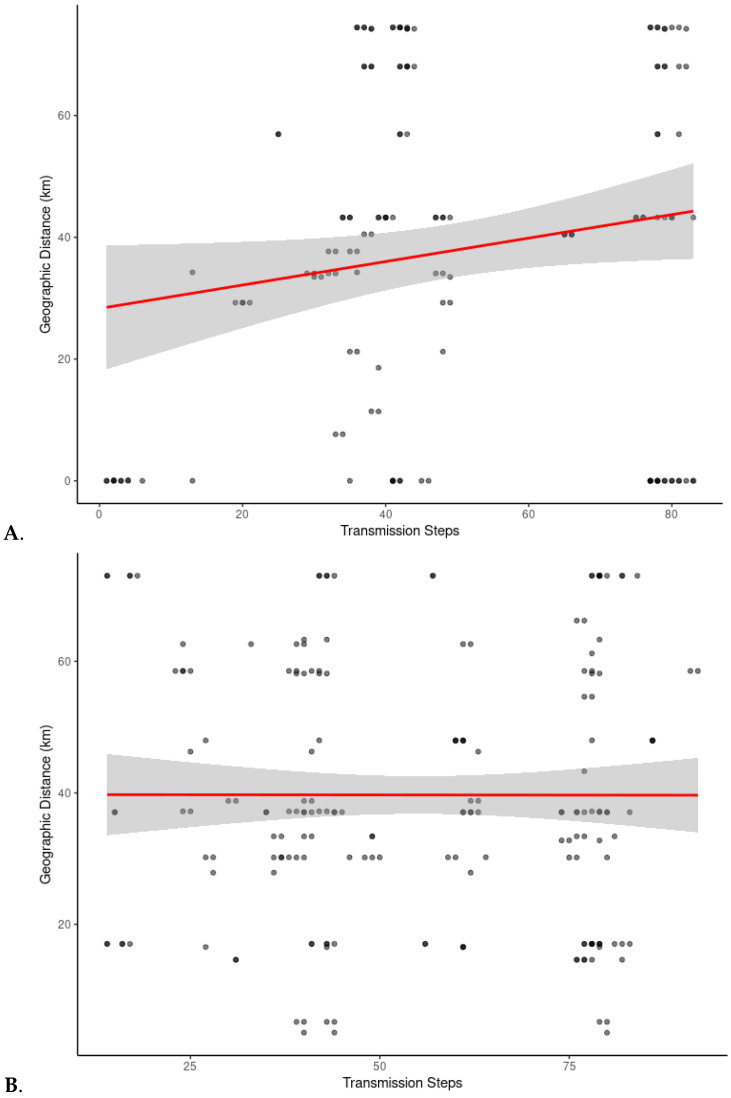
Relationship between geographic distance and inferred transmission steps for (**A**) wild–wild pig pairs and (**B**) wild–farmed pig pairs. Each point represents a pair of individuals within the same clade (Clade 1 or Clade 3). Transmission steps were derived from inferred transmission trees. Geographic distances between sampling locations were calculated from GPS coordinates. A linear regression curve (red line) was fitted to each plot to illustrate general trends. Shaded areas represent the 95% confidence intervals. Darker dots represent overlapping data points, while single or less-overlapped points appear as gray dots.

**Table 1 microorganisms-13-02751-t001:** Summary of transmission steps among different types of pairwise comparisons within two major clades of the phylogenetic tree. By observing the clustering of number of transmission steps in the histograms, we classified the transmission steps into four categories: direct transmission (1 step), few transmission steps (2–5 steps), moderate transmission steps (6–10 steps), and numerous transmission steps (>10 steps).

Clade	Type of Pairs	Median [Q1, Q3 ^$^] of TS *	Minimum # of TS	# of Direct Transmission	# of Few TS (2~5)	# of Moderate TS (6~10)	# of Numerous TS (>10)
1	Wild–wild	41 [38, 77]	1	2/136 (1.5%)	5/136 (3.7%)	0	129/136 (94.8%)
	Farm–farm	67 [41, 79]	2	0	3/91 (3.3%)	5/91 (5.5%)	83/91 (91.2%)
	Wild–farmed	59 [39, 78]	13	0	0	0	238 (100%)
3	Wild–wild	65 [6, 66]	2	0	3/10 (30%)	0	7/10 (70%)
	Farm–farm	91 [11, 158]	1	3/153 (2.0%)	16/153 (10.5%)	19/153 (12.4%)	115/153 (75.2%)
	Wild–farmed	91 [61, 96]	23	0	0	0	90 (100%)

^$^ Lower quartile (Q1) and upper quartile (Q3). * Transmission steps (TS) defined as the number of transmission events in the transmission history between a pair of animals. # Number.

## Data Availability

The original contributions presented in this study are included in the article/Appendix A. Further inquiries can be directed to the corresponding author.

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
