# Peer review of "Transmission Dynamics of Torque Teno Sus Virus 1 (TTSuV1) Between Wild and Farmed Pigs: A Molecular Tool for Monitoring Cross-Population Spillover in Swine"

_microorganisms, 2025, doi:10.3390/microorganisms13122751_

Round 1

Reviewer 1 Report

Comments and Suggestions for Authors

Comments:
The manuscript by Li et al. presented an innovative application of molecular epidemiology to investigate the transmission dynamics of TTSuV1 between invasive wild pigs and farmed domestic pigs in Arkansas, USA. This study integrated field sampling, genomic sequencing, and Bayesian transmission inference to quantify pairwise transmission steps as a proxy for population connectivity and biosecurity efficacy. Key findings of this manuscript include higher TTSuV1 prevalence in farmed pigs compared to wild boars, distribution across four subtypes, and evidence of strong isolation between populations, supporting TTSuV1's potential as a surveillance marker for spillover events. This is a good study; however, I still have concerns to be addressed.

Major concerns:

  1. The use of TransPhylo is innovative for reconstructing transmission chains in a non-pathogenic virus, but key parameters are proxies without direct validation for TTSuV1. The generation time (gamma-distributed mean of 18 days, borrowed from PCV2) assumes similar transmission biology between two ssDNA viruses, yet TTSuV1's persistence and co-infection associations [24–26] may alter infectious periods. Please discuss more about this weak point.
  2. Pairwise step quantification is a clever metric for barriers, but the histograms and heatmaps need more quantitative interpretation—e.g., median/IQR steps for wild-wild vs. farm-farm vs. wild-farmed pairs.

Minor concerns:

  1. Replace Figure 5 with a higher resolution one.
  2. Suggest adding a table for prevalence by farm/site and a summary of regression coefficients (e.g., incidence rate ratios with CIs).

Author Response

Comments from Reviewer 1:

The manuscript by Li et al. presented an innovative application of molecular epidemiology to investigate the transmission dynamics of TTSuV1 between invasive wild pigs and farmed domestic pigs in Arkansas, USA. This study integrated field sampling, genomic sequencing, and Bayesian transmission inference to quantify pairwise transmission steps as a proxy for population connectivity and biosecurity efficacy. Key findings of this manuscript include higher TTSuV1 prevalence in farmed pigs compared to wild boars, distribution across four subtypes, and evidence of strong isolation between populations, supporting TTSuV1's potential as a surveillance marker for spillover events. This is a good study; however, I still have concerns to be addressed.

Response: Thank you for your positive and thoughtful comments on our manuscript. We have addressed all points that you raised in detail below.

Major concerns:

  1. The use of TransPhylo is innovative for reconstructing transmission chains in a non-pathogenic virus, but key parameters are proxies without direct validation for TTSuV1. The generation time (gamma-distributed mean of 18 days, borrowed from PCV2) assumes similar transmission biology between two ssDNA viruses, yet TTSuV1's persistence and co-infection associations [24–26] may alter infectious periods. Please discuss more about this weak point.

Response: We thank the reviewer for this insightful comment. We agree that the generation time parameter used in TransPhylo represents an important source of uncertainty, as no empirical estimates are currently available for TTSuV1. In our analysis, we used the PCV2-based gamma distribution as a biologically reasonable proxy given both are small, circular ssDNA viruses that can establish persistent infections. However, we acknowledge that TTSuV1’s transmission dynamics may differ substantially due to its typically subclinical nature, ubiquitous presence, and frequent co-infection with other pathogens, all of which could lead to longer or more variable infectious periods.

In the revised manuscript, we have expanded our Discussion to emphasize that, while this assumption introduces uncertainty in the absolute timing and number of inferred transmission events, our results reliably capture the relative transmission relationships among hosts. A longer assumed generation time would simply compress the apparent number of transmission steps, while a shorter one would expand them. Thus, differences in parameter choice are expected to shift the numerical cutoffs defining “few”, “moderate”, and “numerous” transmission steps, but not to alter the overall topology or relative connectivity patterns observed in our inferred transmission networks. We also stress the need for empirical studies of TTSuV1 shedding dynamics to refine future model parameterization.

From the revised discussion:

“Second, we inferred the transmission trees using a generation time distribution based on PCV2, due to the lack of published estimates for TTSuV1. While PCV2 is a biologically similar virus, we acknowledge that TTSuV1’s persistent, typically subclinical infection profile may involve longer or more variable transmission intervals. This limitation primarily affects the absolute temporal scaling of our inferred transmission trees, but not their relative structure. In other words, our results reliably capture the relative transmission relationships among hosts, whereas the exact timing and number of inferred transmission events depend on the assumed generation time. A longer generation time would compress the apparent number of transmission steps, and a shorter one would expand them, effectively shifting the boundaries that distinguish few, moderate, and numerous transmission step categories without changing their relative connectivity patterns. Therefore, our interpretations focus on comparative patterns of transmission rather than exact temporal estimates. Future empirical studies characterizing TTSuV1 shedding duration and transmission rates will be essential to refine these parameters and improve the quantitative resolution of transmission inference.”

  1. Pairwise step quantification is a clever metric for barriers, but the histograms and heatmaps need more quantitative interpretation—e.g., median/IQR steps for wild-wild vs. farm-farm vs. wild-farmed pairs.

Response: We agree that providing summary statistics for the pairwise step distances would help readers easily understand the overall distribution of transmission steps in each pair type and help contextualize the patterns shown in the histograms and heatmaps. Accordingly, we now report the median and interquartile range (IQR) for each category of pair type in the revised manuscript and have incorporated these values into Table 1.

Minor concerns:

  1. Replace Figure 5 with a higher resolution one.

Response: We thank the reviewer for this suggestion. Figure 5 has been replaced with a higher-resolution version in the revised manuscript.

  1. Suggest adding a table for prevalence by farm/site and a summary of regression coefficients (e.g., incidence rate ratios with CIs).

Response: In the revised manuscript, we have added a supplementary table (Table S1) summarizing TTSuV1 prevalence by farm and a table (Table S4) summarizing regression coefficients, including incidence rate ratios (IRRs) with 95% confidence intervals.

Reviewer 2 Report

Comments and Suggestions for Authors

The article entitled “Transmission Dynamics of Torque Teno Sus Virus 1 (TTSuV1) Between Wild and Farmed Pigs: A Molecular Tool for Monitoring Cross-Population Spillover in Swine” by Xiaolong Li and coll., is a well written and a very interesting paper. The authors, by combining genetic data with phylogenetic, epidemiological, and spatial analyses, showed that TTSuV1 circulation in Arkansas (USA) is currently structured within wild and farmed pig populations, with no evidence of recent cross-population spillover and spread. TTSuV1 persistence, high prevalence, and genetic variability support its use as a stable, informative marker for surveillance purposes. The huge statistical work has been appreciated.

This manuscript is ready to be accepted, should the authors consider either some major or minor comments.

Major revisions:

Introduction: the first part before line 71 should be shortened (or some information moved to Discussion) (this is only an opinion and not a mandatory action).

LINE 75: Even if this paper is purely concerning veterinary epidemiology, the authors could dedicate a couple of sentences to the diseases TTV can cause in humans. See for example: Righi, F. et al. Torque Teno Sus Virus (TTSuV) Prevalence in Wild Fauna of Northern Italy. Microorganisms 2022, 10, 242. https://doi.org/10.3390/microorganisms10020242.

Suggestion: Righi et al., 2022 should also fit in with the references list cited at line 431.

LINE 90-91: “Without this …”. The authors should better explain this sentence: why should a practical use be theoretical?

LINE 435: I agree with this statement even if the main part of farms in Arkansas is middle-sized.

LINE 562: is there any reference in literature studying the spread/transmission of TTV by birds or small rodents, affecting the farm’s biosecurity?

Minor revisions:

LINE 16: … Arkansas (USA).

LINE 123: What is for “ul”? Do the authors mean “µL”?

LINE 128: separate 15,000 from g

LINE 131: separate 24 from h and edit “to digest protein” to “to digest proteins”.

LINE 132:  … ”Following proteins digestion …”

Author Response

Comments from Reviewer 2:

The article entitled “Transmission Dynamics of Torque Teno Sus Virus 1 (TTSuV1) Between Wild and Farmed Pigs: A Molecular Tool for Monitoring Cross-Population Spillover in Swine” by Xiaolong Li and coll., is a well written and a very interesting paper. The authors, by combining genetic data with phylogenetic, epidemiological, and spatial analyses, showed that TTSuV1 circulation in Arkansas (USA) is currently structured within wild and farmed pig populations, with no evidence of recent cross-population spillover and spread. TTSuV1 persistence, high prevalence, and genetic variability support its use as a stable, informative marker for surveillance purposes. The huge statistical work has been appreciated.

This manuscript is ready to be accepted, should the authors consider either some major or minor comments.

Response: Thank you for your positive and thoughtful comments on our manuscript. We have addressed all points that you raised in detail below.

Major revisions:

  1. Introduction: the first part before line 71 should be shortened (or some information moved to Discussion) (this is only an opinion and not a mandatory action).

Response: We thank the reviewer for this thoughtful suggestion. We appreciate the consideration of shortening the introductory section. However, we have chosen to retain the current structure, as the early introduction of molecular epidemiology concepts and key terms (e.g., pairwise transmission steps, transmission inference) is intentional. These terms are central to our analytical framework and are revisited throughout the Results and Discussion. Presenting them early provides necessary context for readers who may be less familiar with transmission tree reconstruction or quantitative phylodynamic methods, thereby improving the accessibility and logical flow of the manuscript. We believe this approach strengthens the clarity of subsequent sections without substantially lengthening the introduction.

  1. LINE 75: Even if this paper is purely concerning veterinary epidemiology, the authors could dedicate a couple of sentences to the diseases TTV can cause in humans. See for example: Righi, F. et al.Torque Teno Sus Virus (TTSuV) Prevalence in Wild Fauna of Northern Italy. Microorganisms 2022, 10, 242. https://doi.org/10.3390/microorganisms10020242.

Suggestion: Righi et al., 2022 should also fit in with the references list cited at line 431.

Response: We have added a brief overview of TTVs in humans and their broader host distribution. This provides context for TTSuV1 within the Anelloviridae family and acknowledges the relevance of TTVs to both human and animal health, as suggested. And the reference to Righi et al. (2022) has been included in this paragraph. Given the content at line 431 concerns our hypothesis about transmission barriers, we believe the most appropriate placement for this citation is in the introductory section.

From the revised introduction:

“Torque teno viruses (TTVs), a group of non-enveloped, single-stranded DNA viruses belonging to the family Anelloviridae, were first discovered in a human patient in 1997 [23]. Although usually associated with persistent, subclinical infections, human TTVs have been linked to various clinical conditions, including respiratory diseases, acute enteritis, viral hepatitis, and autoimmune rheumatic disorders [24]. Homologous viruses have since been identified in a wide range of domestic and wild animal species, including swine, where they are known as torque teno sus viruses (TTSuVs). TTSuVs are classified into 2 genera, and TTSuV1 belongs to genus Iotatorqueviruses [25].”

  1. LINE 90-91: “Without this …”. The authors should better explain this sentence: why should a practical use be theoretical?

Response: We have clarified this sentence to better explain our intended meaning. Our point was to emphasize that, although TTSuV1 has potential practical value as a marker for viral movement within/between populations, the absence of empirical data on its transmission dynamics prevents this potential from being applied in practice. The sentence has been revised as follows:

“Without empirical data on TTSuV1 transmission dynamics, its potential use as an indicator of viral movement or interpopulation contact in surveillance and risk assessment remains hypothetical.”

  1. LINE 435: I agree with this statement even if the main part of farms in Arkansas is middle-sized.

Response: We thank the reviewer for this positive remark and appreciate their agreement with our interpretation.

  1. LINE 562: is there any reference in literature studying the spread/transmission of TTV by birds or small rodents, affecting the farm’s biosecurity?

Response: We thank the reviewer for raising this point. To our knowledge, there are no published studies documenting the spread or transmission of TTV/anelloviruses by birds or small rodents, nor any evidence that such transmission affects pig farm biosecurity.

Minor revisions:

LINE 16: … Arkansas (USA).

LINE 123: What is for “ul”? Do the authors mean “µL”?

LINE 128: separate 15,000 from g

LINE 131: separate 24 from h and edit “to digest protein” to “to digest proteins”.

LINE 132:  … ”Following proteins digestion …”

Response: We have made these changes in the revised manuscript accordingly.

Round 2

Reviewer 1 Report

Comments and Suggestions for Authors

No further comment. 

Reviewer 2 Report

Comments and Suggestions for Authors

No more comments